# Tractable Operations for Arithmetic Circuits of Probabilistic Models

**Yujia Shen** and **Arthur Choi** and **Adnan Darwiche**
Computer Science Department
University of California
Los Angeles, CA 90095
{yujias,aychoi,darwiche}@cs.ucla.edu

## Abstract

We consider tractable representations of probability distributions and the polytime operations they support. In particular, we consider a recently proposed arithmetic circuit representation, the Probabilistic Sentential Decision Diagram (PSDD). We show that PSDDs support a polytime multiplication operator, while they do not support a polytime operator for summing-out variables. A polytime multiplication operator makes PSDDs suitable for a broader class of applications compared to classes of arithmetic circuits that do not support multiplication. As one example, we show that PSDD multiplication leads to a very simple but effective compilation algorithm for probabilistic graphical models: represent each model factor as a PSDD, and then multiply them.

## 1 Introduction

Arithmetic circuits (ACs) have been a central representation for probabilistic graphical models, such as Bayesian networks and Markov networks. On the reasoning side, some state-of-the-art approaches for exact inference are based on compiling probabilistic graphical models into arithmetic circuits [Darwiche, 2003]; see also Darwiche [2009, chapter 12]. Such approaches can exploit parametric structure (such as determinism and context-specific independence), allowing inference to scale sometimes to models with very high treewidth, which are beyond the scope of classical inference algorithms such as variable elimination and jointree. For example, the ace system for compiling ACs [Chavira and Darwiche, 2008] was the only system in the UAI'08 evaluation of probabilistic reasoning systems to exactly solve all 250 networks in a challenging (very high-treewidth) suite of relational models [Darwiche et al., 2008].

On the learning side, arithmetic circuits have become a popular representation for learning from data, as they are tractable for certain probabilistic queries. For example, there are algorithms for learning ACs of Bayesian networks [Lowd and Domingos, 2008], ACs of Markov networks [Lowd and Rooshenas, 2013, Bekker et al., 2015] and Sum-Product Networks (SPNs) [Poon and Domingos, 2011], among other related representations.[1]

Depending on their properties, different classes of ACs are tractable for different queries and operations. Among these queries are maximum a posteriori (MAP) inference,[2] which is an NP-complete problem, and evaluating the partition function, which is a PP-complete problem (more intractable). Among operations, the multiplication of two ACs stands out as particularly important, being a primitive operation in some approaches to incremental or adaptive inference [Delcher et al., 1995, Acar et al., 2008], bottom-up compilation of probabilistic graphical models [Choi et al., 2013], and some search-based approaches to structure learning [Bekker et al., 2015].

In this paper, we investigate the tractability of two fundamental operations on arithmetic circuits: multiplying two ACs and summing out a variable from an AC. We show that both operations are intractable for some influential ACs that have been employed in the probabilistic reasoning and learning literatures. We then consider a recently proposed sub-class of ACs, called the Probabilistic Sentential Decision Diagram (PSDD) [Kisa et al., 2014]. We show that PSDDs support a polytime multiplication operation, which makes them suitable for a broader class of applications. We also show that PSDDs do not support a polytime summing-out operation (a primitive operation for message-passing inference algorithms). We empirically illustrate the advantages of PSDDs compared to other AC representations, for compiling probabilistic graphical models. Previous approaches for compiling probabilistic models into ACs are based on encoding these models into auxiliary logical representations, such as a Sentential Decision Diagram (SDD) or a deterministic DNNF circuits, which are then converted to an AC [Chavira and Darwiche, 2008, Choi et al., 2013]. PSDDs are a direct representation of probability distributions, bypassing the overhead of intermediate logical representations, and leading to more efficient compilations in some cases. Most importantly though, this approach lends itself to a significantly simpler compilation algorithm: represent each factor of a given model as a PSDD, and then multiply the factors using PSDD multiplication.

This paper is organized as follows. In Section 2, we review arithmetic circuits (ACs) as a representation of probability distributions, including PSDDs in particular. In Section 3, we introduce a polytime multiplication operator for PSDDs, and in Section 4, we show that there is no polytime sum-out operator for PSDDs. In Section 5, we propose a simple compilation algorithm for PSDDs based on the multiply operator, which we evaluate empirically. We discuss related work in Section 6 and finally conclude in Section 7. Proofs of theorems are available in the Appendix.

## 2 Representing Distributions Using Arithmetic Circuits

We start with the definition of factors, which include distributions as a special case.

**Definition 1 (Factor)** *A* factor $f(\mathbf{X})$ *over variables* $\mathbf{X}$ *maps each instantiation* $\mathbf{x}$ *of variables* $\mathbf{X}$ *into a non-negative number* $f(\mathbf{x})$. *The factor represents a distribution when* $\sum_{\mathbf{x}} f(\mathbf{x}) = 1$.

We define the value of a factor at a partial instantiation $\mathbf{y}$, where $\mathbf{Y} \subseteq \mathbf{X}$, as $f(\mathbf{y}) = \sum_{\mathbf{z}} f(\mathbf{yz})$, where $\mathbf{Z} = \mathbf{X} \setminus \mathbf{Y}$. When the factor is a distribution, $f(\mathbf{y})$ corresponds to the probability of evidence $\mathbf{y}$. We also define the MAP instantiation of a factor as $\operatorname{argmax}_{\mathbf{x}} f(\mathbf{x})$, which corresponds to the most likely instantiation when the factor is a distribution.

The classical, tabular representation of a factor $f(\mathbf{X})$ is exponential in the number of variables $\mathbf{X}$. However, one can represent such factors more compactly using arithmetic circuits.

**Definition 2 (Arithmetic Circuit)** *An arithmetic circuit* $\mathcal{AC}(\mathbf{X})$ *over variables* $\mathbf{X}$ *is a rooted DAG whose internal nodes are labeled with* $+$ *or* $*$ *and whose leaf nodes are labeled with either indicator variables* $\lambda_x$ *or non-negative parameters* $\theta$. *The value of the circuit at instantiation* $\mathbf{x}$, *denoted* $\mathcal{AC}(\mathbf{x})$, *is obtained by assigning indicator* $\lambda_x$ *the value* $1$ *if* $x$ *is compatible with instantiation* $\mathbf{x}$ *and* $0$ *otherwise, then evaluating the circuit in the standard way. The circuit* $\mathcal{AC}(\mathbf{X})$ *represents factor* $f(\mathbf{X})$ *iff* $\mathcal{AC}(\mathbf{x}) = f(\mathbf{x})$ *for each instantiation* $\mathbf{x}$.

A *tractable* arithmetic circuit allows one to efficiently answer certain queries about the factor it represents. We next discuss two properties that lead to tractable arithmetic circuits. The first is *decomposability* [Darwiche, 2001b], which was used for probabilistic reasoning in [Darwiche, 2003].

**Definition 3 (Decomposability)** *Let* $n$ *be a node in an arithmetic circuit* $\mathcal{AC}(\mathbf{X})$. *The variables of* $n$, *denoted* $vars(n)$, *are the variables* $X \in \mathbf{X}$ *with some indicator* $\lambda_x$ *appearing at or under node* $n$. *An arithmetic circuit is decomposable iff every pair of children* $c_1$ *and* $c_2$ *of a* $*$-*node satisfies* $vars(c_1) \cap vars(c_2) = \emptyset$.

The second property is *determinism* [Darwiche, 2001a], which was also employed for probabilistic reasoning in Darwiche [2003].

**Definition 4 (Determinism)** *An arithmetic circuit* $\mathcal{AC}(\mathbf{X})$ *is deterministic iff each* $+$-*node has at most one non-zero input when the circuit is evaluated under any instantiation* $\mathbf{x}$ *of the variables* $\mathbf{X}$.

A third property called *smoothness* is also desirable as it simplifies the statement of certain AC algorithms, but is less important for tractability as it can be enforced in polytime [Darwiche, 2001a].

**Definition 5 (Smoothness)** *An arithmetic circuit $\mathcal{AC}(\mathbf{X})$ is smooth iff it contains at least one indicator for each variable in $\mathbf{X}$, and for each child $c$ of +-node $n$, we have $vars(n) = vars(c)$.*

Decomposability and determinism lead to tractability in the following sense. Let $Pr(\mathbf{X})$ be a distribution represented by a decomposable, deterministic and smooth arithmetic circuit $\mathcal{AC}(\mathbf{X})$. Then one can compute the following queries in time that is linear in the size of circuit $\mathcal{AC}(\mathbf{X})$: the probability of any partial instantiation, $Pr(\mathbf{y})$, where $\mathbf{Y} \subseteq \mathbf{X}$ [Darwiche, 2003] and the most likely instantiation, $\operatorname{argmax}_{\mathbf{x}} Pr(\mathbf{x})$ [Chan and Darwiche, 2006]. The decision problems of these queries are known to be PP-complete and NP-complete for Bayesian networks [Roth, 1996, Shimony, 1994].

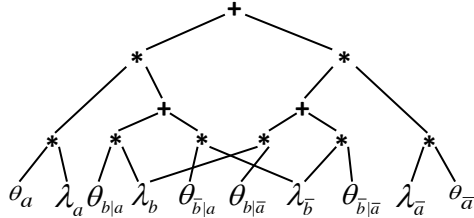

Figure 1: An AC for a Bayesian network $A \to B$.

A number of methods have been proposed for compiling a Bayesian network into a decomposable, deterministic and smooth AC that represents its distribution [Darwiche, 2003]. Figure 1 depicts such a circuit that represents the distribution of Bayesian network $A \to B$. One method ensures that the size of the AC is proportional to the size of a jointree for the network. Another method yields circuits that can sometimes be exponentially smaller, and is implemented in the publicly available ace system [Chavira and Darwiche, 2008]; see also Darwiche et al. [2008]. Additional methods are discussed in Darwiche [2009, chapter 12].

This work is motivated by the following limitation of these tractable circuits, which may narrow their applicability in probabilistic reasoning and learning.

**Definition 6 (Multiplication)** *The product of two arithmetic circuits $\mathcal{AC}_1(\mathbf{X})$ and $\mathcal{AC}_2(\mathbf{X})$ is an arithmetic circuit $\mathcal{AC}(\mathbf{X})$ such that $\mathcal{AC}(\mathbf{x}) = \mathcal{AC}_1(\mathbf{x})\mathcal{AC}_2(\mathbf{x})$ for every instantiation $\mathbf{x}$.*

**Theorem 1** *Computing the product of two decomposable ACs is NP-hard if the product is also decomposable. Computing the product of two decomposable and deterministic ACs is NP-hard if the product is also decomposable and deterministic.*

We now investigate a newly introduced class of tractable ACs, called the Probabilistic Sentential Decision Diagram (PSDD) [Kisa et al., 2014]. In particular, we show that this class of circuits admits a tractable product operation and then explore an application of this operation to exact inference in probabilistic graphical models.

PSDDs were motivated by the need to represent probability distributions $Pr(\mathbf{X})$ with many instantiations $\mathbf{x}$ attaining zero probability, $Pr(\mathbf{x}) = 0$. Consider the distribution $Pr(\mathbf{X})$ in Figure 2(a) for an example. The first step in constructing a PSDD for this distribution is to construct a special Boolean circuit that captures its zero entries; see Figure 2(b). The Boolean circuit captures zero entries in the following sense. For each instantiation $\mathbf{x}$, the circuit evaluates to 0 at instantiation $\mathbf{x}$ iff $Pr(\mathbf{x}) = 0$. The second and final step of constructing a PSDD amounts to parameterizing this Boolean circuit (e.g., by learning them from data), by including a local distribution on the inputs of each or-gate; see Figure 2(c).

The Boolean circuit underlying a PSDD is known as a Sentential Decision Diagram (SDD) [Darwiche, 2011]. These circuits satisfy specific syntactic and semantic properties based on a binary tree, called a *vtree,* whose leaves correspond to variables; see Figure 2(d). The following definition of SDD circuits is a based on the one given by Darwiche [2011] and uses a different notation.

**Definition 7 (SDD)** *An SDD normalized for a vtree $v$ is a Boolean circuit defined as follows. If $v$ is a leaf node labeled with variable $X$, then the SDD is either $X$, $\neg X$, $\bot$ or an or-gate with inputs $X$ and $\neg X$. If $v$ is an internal vtree node, then the SDD has the structure in Figure 3, where $p_1, \ldots, p_n$ are SDDs normalized for the left child $v^l$ and $s_1, \ldots, s_n$ are SDDs normalized for the right child $v^r$. Moreover, the circuits $p_1, \ldots, p_n$ are consistent, mutually exclusive and exhaustive.*

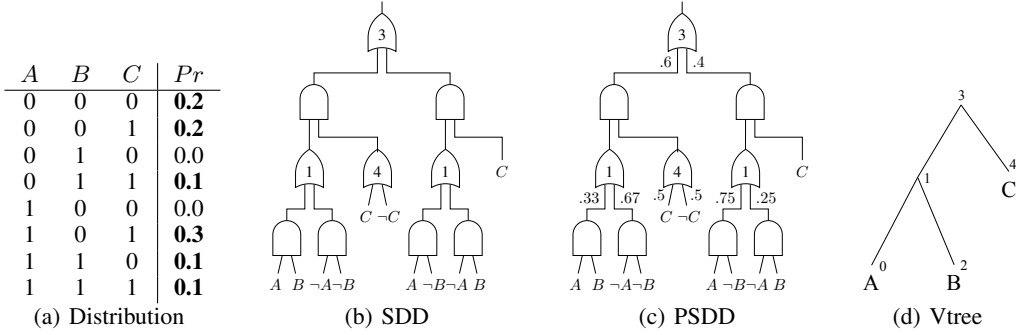

| A | B | C | Pr |
|---|---|---|-----|
| 0 | 0 | 0 | **0.2** |
| 0 | 0 | 1 | **0.2** |
| 0 | 1 | 0 | 0.0 |
| 0 | 1 | 1 | **0.1** |
| 1 | 0 | 0 | 0.0 |
| 1 | 0 | 1 | **0.3** |
| 1 | 1 | 0 | **0.1** |
| 1 | 1 | 1 | **0.1** |

(a) Distribution      (b) SDD      (c) PSDD      (d) Vtree

Figure 2: A probability distribution and its SDD/PSDD representation. Note that the numbers annotating or-gates in (b) & (c) correspond to vtree node IDs in (d). Further, note that while the circuit appears to be a tree, the input variables are shared and hence the circuit is not a tree.

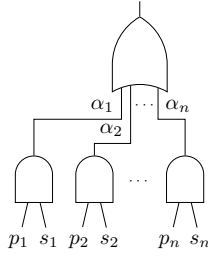

Figure 3: Each $(p_i, s_i, \alpha_i)$ is called an element of the or-gate, where the $p_i$'s are called primes and the $s_i$'s are called subs. Moreover, $\sum_i \alpha_i = 1$ and exactly one $p_i$ evaluates to 1 under any circuit input.

SDD circuits alternate between or-gates and and-gates. Their and-gates have two inputs each. The or-gates of these circuits are such that at most one input will be high under any circuit input. An SDD circuit may produce a 1-output for every possible input (i.e., the circuit represents the function true). These circuits arise when representing strictly positive distributions (with no zero entries).

A PSDD is obtained by including a distribution $\alpha_1, \ldots, \alpha_n$ on the inputs of each or-gate; see Figure 3. The semantics of PSDDs are given in [Kisa et al., 2014].[3] We next provide an alternative semantics, which is based on converting a PSDD into an arithmetic circuit.

**Definition 8 (ACs of PSDDs)** *The arithmetic circuit of a PSDD is obtained as follows. Leaf nodes $x$ and $\perp$ are converted into $\lambda_x$ and $0$, respectively. Each and-gate is converted into a $*$-node. Each or-node with children $c_1, \ldots, c_n$ and corresponding parameters $\alpha_1, \ldots, \alpha_n$ is converted into a $+$-node with children $\alpha_1 * c_1, \ldots, \alpha_n * c_n$.*

**Theorem 2** *The arithmetic circuit of a PSDD represents the distribution induced by the PSDD. Moreover, the arithmetic circuit is decomposable and deterministic.*[4]

The PSDD is a complete and canonical representation of probability distributions. That is, PSDDs can represent any distribution, and there is a unique PSDD for that distribution (under some conditions). A variety of probabilistic queries are tractable on PSDDs, including that of computing the probability of a partial variable instantiation and the most likely instantiation. Moreover, the maximum likelihood parameter estimates of a PSDD are unique given complete data, and these parameters can be computed efficiently using closed-form estimates; see [Kisa et al., 2014] for details. Finally, PSDDs have been used to learn distributions over *combinatorial objects,* including rankings and permutations [Choi et al., 2015], paths and games [Choi et al., 2016]. In these applications, the Boolean circuit underlying a PSDD captures variable instantiations that correspond to combinatorial objects, while its parameterization induces a distribution over these objects.

As a concrete example, PSDDs were used to induce distributions over the permutations of $n$ items as follows. We have a variable $X_{ij}$ for each $i, j \in \{1, \ldots, n\}$ denoting that item $i$ is at position $j$ in the permutation. Clearly, not all instantiations of these variables correspond to (valid) permutations. An SDD circuit is then constructed, which outputs 1 iff the corresponding input corresponds to a valid permutation. Each parameterization of this SDD circuit leads to a distribution on permutations and these parameterizations can be learned from data; see Choi et al. [2015].

# 3 Multiplying Two PSDDs

Factors and their operations are fundamental to probabilistic inference, whether exact or approximate [Darwiche, 2009, Koller and Friedman, 2009]. Consider two of the most basic operations on factors: (1) computing the product of two factors and (2) summing out a variable from a factor. With these operations, one can directly implement various inference algorithms, including variable elimination, the jointree algorithm, and message-passing algorithms such as loopy belief propagation. Typically, tabular representations (and their sparse variations) are used to represent factors and implement the above algorithms; see Larkin and Dechter [2003], Sanner and McAllester [2005], Chavira and Darwiche [2007] for some alternatives.

More generally, factor multiplication is useful for online or incremental reasoning with probabilistic models. In some applications, we may not have access to all factors of a model beforehand, to compile as a jointree or an arithmetic circuit. For example, when learning the structure of a Markov network from data [Bekker et al., 2015], we may want to introduce and remove candidate factors from a model, while evaluating the changes to the log likelihood. Certain realizations of generalized belief propagation also require the multiplication of factors [Yedidia et al., 2005, Choi and Darwiche, 2011]. In these realizations, one can use factor multiplication to enforce dependencies between factors that have been relaxed to make inference more tractable, albeit less accurate.

We next discuss PSDD multiplication, while deferring summing out to the following section.

---

**Algorithm 1** Multiply$(n_1, n_2, v)$

---

**input:** PSDDs $n_1, n_2$ normalized for vtree $v$

**output:** PSDD $n$ and constant $\kappa$

**main:**
1:   $n, k \leftarrow \mathsf{cache_m}(n_1, n_2), \mathsf{cache_c}(n_1, n_2)$            ▷ check if previously computed
2:   **if** $n \neq$ null **then return** $(n, k)$            ▷ return previously cached result
3:   **else if** $v$ is a leaf **then** $(n, \kappa) \leftarrow \mathsf{BaseCase}(n_1, n_2)$        ▷ $n_1, n_2$ are literals, $\bot$ or simple or-gates
4:   **else**            ▷ $n_1$ and $n_2$ have the structure in Figure 3
5:      $\gamma, \kappa \leftarrow \{\}, 0$            ▷ initialization
6:      **for all** elements $(p, s, \alpha)$ of $n_1$ **do**            ▷ see Figure 3
7:         **for all** elements $(q, r, \beta)$ of $n_2$ **do**            ▷ see Figure 3
8:            $(m_1, k_1) \leftarrow \mathsf{Multiply}(p, q, v^l)$        ▷ recursively multiply primes $p$ and $q$
9:            **if** $k_1 \neq 0$ **then**            ▷ if $(m_1, k_1)$ is not a trivial factor
10:               $(m_2, k_2) \leftarrow \mathsf{Multiply}(s, r, v^r)$        ▷ recursively multiply subs $s$ and $r$
11:               $\eta \leftarrow k_1 \cdot k_2 \cdot \alpha \cdot \beta$        ▷ compute weight of element $(m_1, m_2)$
12:               $\kappa \leftarrow \kappa + \eta$        ▷ aggregate weights of elements
13:               add $(m_1, m_2, \eta)$ to $\gamma$
14:      $\gamma \leftarrow \{(m_1, m_2, \eta/\kappa) \mid (m_1, m_2, \eta) \in \gamma\}$        ▷ normalize parameters of $\gamma$
15:      $n \leftarrow$ unique PSDD node with elements $\gamma$        ▷ cache lookup for unique nodes
16: $\mathsf{cache_m}(n_1, n_2) \leftarrow n$
17: $\mathsf{cache_c}(n_1, n_2) \leftarrow \kappa$        ▷ store results in cache
18: **return** $(n, \kappa)$

---

Our first observation is that the product of two distributions is generally not a distribution, but a factor. Moreover, a factor $f(\mathbf{X})$ can always be represented by a distribution $Pr(\mathbf{X})$ and a constant $\kappa$ such that $f(\mathbf{x}) = \kappa \cdot Pr(\mathbf{x})$. Hence, our proposed multiplication method will output a PSDD together with a constant, as given in Algorithm 1. This algorithm uses three caches, one for storing constants ($\mathsf{cache_c}$), another for storing circuits ($\mathsf{cache_m}$), and a third used to implement Line 15.[5] This line ensures that the PSDD has no duplicate structures of the form given in Figure 3. The description of function $\mathsf{BaseCase}()$ on Line 3 is available in the Appendix. It appears inside the proof of the following theorem, which establishes the soundness and complexity of the given algorithm.

**Theorem 3** *Algorithm 1 outputs a PSDD $n$ normalized for vtree $v$. Moreover, if $Pr_1(\mathbf{X})$ and $Pr_2(\mathbf{X})$ are the distributions of input PSDDs $n_1$ and $n_2$, and $Pr(\mathbf{X})$ is the distribution of output PSDD $n$, then $Pr_1(\mathbf{x})Pr_2(\mathbf{x}) = \kappa \cdot Pr(\mathbf{x})$ for every instantiation $\mathbf{x}$. Finally, Algorithm 1 takes time $O(s_1 s_2)$, where $s_1$ and $s_2$ are the sizes of input PSDDs.*

We will later discuss an application of PSDD multiplication to probabilistic inference, in which we cascade these multiplication operations. In particular, we end up multiplying two factors $f_1$ and $f_2$, represented by PSDDs $n_1$ and $n_2$ and the corresponding constants $\kappa_1$ and $\kappa_2$. We use Algorithm 1 for this purpose, multiplying PSDDs $n_1$ and $n_2$ (distributions), to yield a PSDD $n$ (distribution) and a constant $\kappa$. The factor $f_1 f_2$ will then correspond to PSDD $n$ and constant $\kappa \cdot \kappa_1 \cdot \kappa_2$.

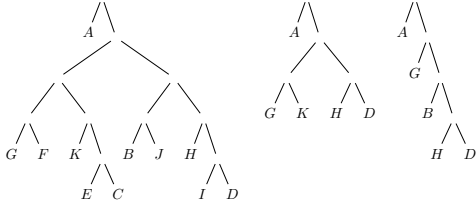

Figure 4: A vtree and two of its projections.

Another observation is that Algorithm 1 assumes that the input PSDDs are over the same vtree and, hence, same set of variables. A more detailed version of this algorithm can multiply two PSDDs over different sets of variables as long as the PSDDs have *compatible* vtrees. We omit this version here to simplify the presentation, but mention that it has the same complexity as Algorithm 1.

Two vtrees over variables $\mathbf{X}$ and $\mathbf{Y}$ are compatible iff they can be obtained by projecting some vtree on variables $\mathbf{X}$ and $\mathbf{Y}$, respectively.

**Definition 9 (Vtree Projection)** *Let $v$ be a vtree over variables $\mathbf{Z}$. The projection of $v$ on variables $\mathbf{X} \subseteq \mathbf{Z}$ is obtained as follows. Successively remove every maximal subtree $v'$ whose variables are outside $\mathbf{X}$, while replacing the parent of $v'$ with its sibling.*

Figure 4 depicts a vtree and two of its projections. When compiling a probabilistic graphical model into a PSDD, we first construct a vtree $v$ over all variables in the model. We then compile each factor $f(\mathbf{X})$ into a PSDD, using the projection of $v$ on variables $\mathbf{X}$. We finally multiply the PSDDs of these factors. We will revisit these steps later.

## 4 Summing-Out a Variable in a PSDD

We now discuss the summing out of variables from distributions represented by arithmetic circuits.

**Definition 10 (Sum Out)** *Summing-out a variable $X \in \mathbf{X}$ from factor $f(\mathbf{X})$ results in another factor over variables $\mathbf{Y} = \mathbf{X} \setminus \{X\}$, denoted by $\sum_X f$ and defined as: $\left( \sum_X f \right)(\mathbf{y}) \overset{def}{=} \sum_x f(x, \mathbf{y})$.*

When the factor is a distribution (i.e., normalized), the sum out operation corresponds to marginalization. Together with multiplication, summing out provides a direct implementation of algorithms such as variable elimination and those based on message passing.

Just like multiplication, summing out is also intractable for a common class of arithmetic circuits.

**Theorem 4** *The sum-out operation on decomposable and deterministic ACs is NP-hard, assuming the output is also decomposable and deterministic.*

This theorem does not preclude the possibility that the resulting AC is of polynomial size with respect to the size of the input AC—it just says that the computation is intractable. Summing out is also intractable on PSDDs, but the result is stronger here as the size of the output can be exponential.

**Theorem 5** *There exists a class of factors $f(\mathbf{X})$ and variable $X \in \mathbf{X}$, such that $n = |\mathbf{X}|$ can be arbitrarily large, $f(\mathbf{X})$ has a PSDD whose size is linear in $n$, while the PSDD of $\sum_X f$ has size exponential in $n$ for every vtree.*

Only the multiplication operation is needed to compile probabilistic graphical models into arithmetic circuits. Even for inference algorithms that require summing out variables, such as variable elimination, summing out can still be useful, even if intractable, since the size of resulting arithmetic circuit will not be larger than a tabular representation.

# 5 Compiling Probabilistic Graphical Models into PSDDs

Even though PSDDs form a strict subclass of decomposable and deterministic ACs (and satisfy stronger properties), one can still provide the following classical guarantee on PSDD size.

**Theorem 6** *The interaction graph of factors $f_1(\mathbf{X}_1), \ldots, f_n(\mathbf{X}_n)$ has nodes corresponding to variables $\mathbf{X}_1 \cup \ldots \cup \mathbf{X}_n$ and an edge between two variables iff they appear in the same factor. There is a PSDD for the product $f_1 \ldots f_n$ whose size is $O(m \cdot \exp(w))$, where $m$ is the number of variables and $w$ is its treewidth.*

This theorem provides an upper bound on the size of PSDD compilations for both Bayesian and Markov networks. An analogous guarantee is available for SDD circuits of propositional models, using a special type of vtree known as a *decision vtree* [Oztok and Darwiche, 2014]. We next discuss our experiments, which focused on the compilation of Markov networks using decision vtrees.

To compile a Markov network, we first construct a decision vtree using a known technique.[6] For each factor of the network, we project the vtree on the factor variables, and then compile the factor into a PSDD. This can be done in time linear in the factor size, but we omit the details here. We finally multiply the obtained PSDDs. The order of multiplication is important to the overall efficiency of the compilation approach. The order we used is as follows. We assign each PSDD to the lowest vtree node containing the PSDD variables, and then multiply PSDDs in the order that we encounter them as we traverse the vtree bottom-up (this is analogous to compiling CNFs in Choi et al. [2013]).

Table 1 summarizes our results. We compiled Markov networks into three types of arithmetic circuits. The first compilation (AC1) is to decomposable and deterministic ACs using ace [Chavira and Darwiche, 2008].[7] The second compilation (AC2) is also to decomposable and deterministic ACs, but using the approach proposed in Choi et al. [2013]. The third compilation is to PSDDs as discussed above. The first two approaches are based on reducing the inference problem into a weighted model counting problem. In particular, these approaches encode the network using Boolean expressions, which are compiled to logical representations (d-DNNF or SDD), from which an arithmetic circuit is induced. The systems underlying these approaches are quite complex and are the result of many years of engineering. In contrast, the proposed compilation to PSDDs does not rely on an intermediate representation or additional boxes, such as d-DNNF or SDD compilers.

The benchmarks in Table 1 are from the UAI-14 Inference Competition.[8] We selected all networks over binary variables in the MAR track, and report a network only if at least one approach successfully compiled it (given time and space limits of 30 minutes and 16GB). We report the size (the number of edges) and time spent for each compilation. First, we note that for all benchmarks that compiled to both PSDD and AC2 (based on SDDs), the PSDD size is always smaller. This can be attributed in part to the fact that reductions to weighted model counting represent parameters explicitly as variables, which are retained throughout the compilation process. In contrast, PSDD parameters are annotated on its edges. More interestingly, when we multiply two PSDD factors, the parameters of the inputs may not persist in the output PSDD. That is, the PSDD only maintains enough parameters to represent the resulting distribution, which further explains the size differences.

In the Promedus benchmarks, we also see that in all but 5 cases, the compiled PSDD is smaller than AC1. However, several Grids benchmarks were compilable to AC1, but failed to compile to AC2 or PSDD, given the time and space limits. On the other hand, we were able to compile some of the relational benchmarks to PSDD, which did not compile to AC1 and compiled partially to AC2.

# 6 Related Work

Tabular representations and their sparse variations (e.g., Larkin and Dechter [2003]) are typically used to represent factors for probabilistic inference and learning. Rules and decision trees are more succinct representations for modeling context-specific independence, although they are not much more amenable to exact inference compared to tabular representations [Boutilier et al., 1996, Friedman and Goldszmidt, 1998]. Domain specific representations have been proposed, e.g., in computer vision

Table 1: AC compilation size (number of edges) and time (in seconds)

| network | compilation size | | | compilation time | | |
|---|---|---|---|---|---|---|
| | AC1 | AC2 | psdd | AC1 | AC2 | psdd |
| Alchemy_11 | 12,705,213 | - | 13,715,906 | 130.83 | - | 300.80 |
| Grids_11 | 81,074,816 | - | - | 271.97 | - | - |
| Grids_12 | 232,496 | 457,529 | 201,250 | 0.93 | 1.12 | 1.68 |
| Grids_13 | 81,090,432 | - | - | 273.88 | - | - |
| Grids_14 | 83,186,560 | - | - | 279.12 | - | - |
| Segmentation_11 | 20,895,884 | 41,603,129 | 30,951,708 | 72.39 | 204.54 | 223.60 |
| Segmentation_12 | 15,840,404 | 41,005,721 | 34,368,060 | 51.27 | 209.03 | 283.79 |
| Segmentation_13 | 33,746,511 | 78,028,443 | 33,726,812 | 117.46 | 388.97 | 255.29 |
| Segmentation_14 | 16,965,928 | 48,333,027 | 46,363,820 | 62.31 | 279.19 | 639.07 |
| Segmentation_15 | 29,888,972 | - | 33,866,332 | 107.87 | - | 273.67 |
| Segmentation_16 | 18,799,112 | 54,557,867 | 19,935,308 | 65.64 | 265.07 | 163.38 |
| relational_3 | - | 183,064 | 41,070 | - | 1.21 | 10.43 |
| relational_5 | - | - | 217,696 | - | - | 594.68 |
| Promedus_11 | 67,036 | 174,592 | 30,542 | 6.80 | 1.88 | 2.28 |
| Promedus_12 | 45,119 | 349,916 | 48,814 | 0.91 | 5.81 | 2.46 |
| Promedus_13 | 42,065 | 83,701 | 26,100 | 0.80 | 0.23 | 3.94 |
| Promedus_14 | 2,354,180 | 3,667,740 | 749,528 | 21.64 | 33.27 | 24.90 |
| Promedus_15 | 14,363 | 31,176 | 9,520 | 0.95 | 0.10 | 1.52 |
| Promedus_16 | 45,935 | 154,467 | 29,150 | 1.35 | 0.40 | 2.06 |
| Promedus_17 | 3,336,316 | 9,849,598 | 1,549,170 | 68.08 | 48.47 | 50.22 |

| network | compilation size | | | compilation time | | |
|---|---|---|---|---|---|---|
| | AC1 | AC2 | psdd | AC1 | AC2 | psdd |
| Promedus_18 | 3,006,654 | 762,247 | 539,478 | 20.46 | 18.38 | 21.20 |
| Promedus_19 | 796,928 | 1,171,288 | 977,510 | 6.80 | 25.01 | 68.62 |
| Promedus_20 | 70,422 | 188,322 | 70,492 | 0.96 | 3.24 | 3.46 |
| Promedus_21 | 17,528 | 31,911 | 10,944 | 0.62 | 0.18 | 1.78 |
| Promedus_22 | 26,010 | 39,016 | 33,064 | 0.63 | 0.10 | 1.58 |
| Promedus_23 | 329,669 | 1,473,628 | 317,514 | 3.29 | 17.77 | 10.88 |
| Promedus_24 | 4,774 | 9,085 | 1,960 | 0.45 | 0.05 | 0.80 |
| Promedus_25 | 556,179 | 3,614,581 | 407,974 | 7.66 | 32.90 | 6.78 |
| Promedus_26 | 57,190 | 24,578 | 5,146 | 0.71 | 198.74 | 2.72 |
| Promedus_27 | 33,611 | 52,698 | 19,434 | 0.73 | 0.55 | 1.16 |
| Promedus_28 | 24,049 | 46,364 | 17,084 | 1.04 | 0.30 | 1.59 |
| Promedus_29 | 10,403 | 20,600 | 4,828 | 0.54 | 0.08 | 1.88 |
| Promedus_30 | 9,884 | 21,230 | 6,734 | 0.50 | 0.07 | 1.23 |
| Promedus_31 | 17,977 | 31,754 | 10,842 | 0.57 | 0.12 | 1.96 |
| Promedus_32 | 15,215 | 33,064 | 8,682 | 0.59 | 0.11 | 1.77 |
| Promedus_33 | 10,734 | 18,535 | 4,006 | 0.59 | 0.07 | 1.57 |
| Promedus_34 | 38,113 | 54,214 | 21,398 | 0.87 | 0.78 | 1.78 |
| Promedus_35 | 18,765 | 31,792 | 11,120 | 0.68 | 0.13 | 1.79 |
| Promedus_36 | 19,175 | 31,792 | 11,004 | 1.22 | 0.12 | 1.91 |
| Promedus_37 | 77,088 | 144,664 | 79,210 | 1.49 | 3.50 | 6.15 |
| Promedus_38 | 177,560 | 593,675 | 123,552 | 1.67 | 17.19 | 8.09 |

[Felzenszwalb and Huttenlocher, 2006], to allow for more efficient factor operations. Algebraic Decision Diagrams (ADDs) and Algebraic Sentential Decision Diagrams (ASDDs) can also be used to multiply two factors in polytime [Bahar et al., 1993, Herrmann and de Barros, 2013], but their sizes can grow quickly with repeated multiplications: ADDs have a distinct node for each possible value of a factor/distribution. Since ADDs also support a polytime summing-out operation, ADDs are more commonly used in the context of variable elimination [Sanner and McAllester, 2005, Chavira and Darwiche, 2007], and in message passing algorithms [Gogate and Domingos, 2013]. Probabilistic Decision Graphs (PDGs) and AND/OR Multi-Valued Decision Diagrams (AOMDD) support a polytime multiply operator, and also have treewidth upper bounds when compiling probabilistic graphical models [Jaeger, 2004, Mateescu et al., 2008]. Both PDGs and AOMDDs can be viewed as sub-classes of PSDDs that branch on variables instead of sentences as is the case with PSDDs—this distinction can lead to exponential reductions in size [Xue et al., 2012, Bova, 2016].

## 7 Conclusion

We considered the tractability of multiplication and summing-out operators for arithmetic circuits (ACs), as tractable representations of factors and distributions. We showed that both operations are intractable for deterministic and decomposable ACs (under standard complexity theoretic assumptions). We also showed that for a sub-class of ACs, known as PSDDs, a polytime multiplication operator is supported. Moreover, we showed that PSDDs do not support summing-out in polytime (unconditionally). Finally, we illustrated the utility of PSDD multiplication, providing a relatively simple but effective algorithm for compiling probabilistic graphical models into PSDDs.

**Acknowledgments**

This work was partially supported by NSF grant #IIS-1514253 and ONR grant #N00014-15-1-2339.

## Footnotes

[1]SPNs can be converted into ACs (and vice-versa) with linear size and time [Rooshenas and Lowd, 2014].

[2]This is also known as most probable explanation (MPE) inference [Pearl, 1988].

[3]Let $\mathbf{x}$ be an instantiation of PSDD variables. If the SDD circuit outputs 0 at input $\mathbf{x}$, then $Pr(\mathbf{x}) = 0$. Otherwise, traverse the circuit top-down, visiting the (unique) high input of each visited or-node, and all inputs of each visited and-node. Then $Pr(\mathbf{x})$ is the product of parameters visited during the traversal process.

[4]The arithmetic circuit also satisfies a minor weakening of smoothness with the same effect as smoothness.

[5]The cache key of a PSDD node in Figure 3 is based on the (unique) ID's of nodes $p_i/s_i$ and parameters $\alpha_i$.

[6] We used the minic2d package which is available at `reasoning.cs.ucla.edu/minic2d/`.

[7] The ace system is publicly available at `http://reasoning.cs.ucla.edu/ace/`.

[8] `http://www.hlt.utdallas.edu/~vgogate/uai14-competition/index.html`

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
