[Supplementary Material · ShenChoiDarwiche-supplementary.pdf]

**Supplementary Material**:

**Tractable Operations for Arithmetic Circuits of Probabilistic Models**

This supplementary appendix contains proofs of the results in this paper.

## A  Proofs

**Proof of Theorem 1**  We first observe that for DNNF circuits $f_1$ and $f_2$, checking whether $f_1 \wedge f_2$ is consistent is NP-hard [Darwiche and Marquis, 2002]. We will now reduce this consistency test to multiplying two decomposable ACs. We first convert each DNNF circuit $f_i$ into an arithmetic circuit $\mathcal{AC}_i$ by replacing or-gates with +-nodes, and-gates with *-nodes, inputs $x$ with $\lambda_x$, and true/false with $1/0$. The resulting AC is decomposable and satisfies $\mathcal{AC}_i(\mathbf{x}) = 0$ iff $f_i$ outputs 0 on input $\mathbf{x}$ (i.e., $\mathcal{AC}_i(\mathbf{x}) > 0$ iff $f_i$ outputs 1 on input $\mathbf{x}$). Hence, $f_1 \wedge f_2$ outputs 0 on input $\mathbf{x}$ iff $\mathcal{AC}_1(\mathbf{x}) \cdot \mathcal{AC}_2(\mathbf{x}) = 0$. Therefore, $f_1 \wedge f_2$ is consistent iff there exists an input $\mathbf{x}$ with $(\mathcal{AC}_1 \cdot \mathcal{AC}_2)(\mathbf{x}) > 0$, which holds iff the partition function of $\mathcal{AC}_1 \cdot \mathcal{AC}_2$ is greater than 0. The latter condition can be checked in polytime on decomposable ACs. An analogous argument can be used for the multiplication of decomposable and deterministic ACs, using a consistency test on the conjunction of two d-DNNFs [Darwiche and Marquis, 2002]. $\square$

**Proof of Theorem 2**  That the AC is decomposable and deterministic follows directly from the definition of a PSDD and its underlying SDD. That the AC represents the distribution induced by the PSDD can be shown using an inductive argument on the structure of the PSDD. For the base case, we have a PSDD literal $x$ and AC indicator $\lambda_x$, or a PSDD terminal $\bot$ and AC constant 0, or a PSDD simple or-node and AC $\alpha\lambda_x + (1-\alpha)\lambda_{\neg x}$. The theorem holds for all cases.

For the inductive case, consider an or-node $n$ of the PSDD with elements $(p_i, s_i, \alpha_i)$, and the corresponding +-node $\mathcal{AC}_n$ of the AC which has the form $\sum_i \alpha_i \cdot \mathcal{AC}_{p_i} \cdot \mathcal{AC}_{s_i}$. By induction, we assume the PSDD/AC pairs $p_i/\mathcal{AC}_{p_i}$ and $s_i/\mathcal{AC}_{s_i}$ induce the same distribution, i.e., the value of the PSDD is the same as the value of the AC given the same input. Given input $\mathbf{x}$, at most one element $(p_i, s_i, \alpha_i)$ of the PSDD will have its corresponding SDD wire evaluate to 1 (see Footnote 3). The value of the PSDD given input $\mathbf{x}$ is the product of $\alpha_i$ and the values of $p_i$ and $s_i$. Similarly, the AC has the same non-zero child $\alpha_i \cdot \mathcal{AC}_{p_i} \cdot \mathcal{AC}_{s_i}$ (the others must have value zero by induction). Hence, the distributions of PSDD $n$ and circuit $\mathcal{AC}_n$ are the same. $\square$

**Proof of Theorem 3**  Let input PSDDs $n_1$ and $n_2$ have elements $(p_i, s_i, \alpha_i)$ and $(q_j, r_j, \beta_j)$, respectively. If vtree $v$ is over variables $\mathbf{X}$, then denote the variables of the left and right children $v^l$ and $v^r$ by $\mathbf{X}^l$ and $\mathbf{X}^r$, respectively. Let $Pr_n$ denote the distribution of a PSDD $n$. First, for an instantiation $\mathbf{x}^l$ there is a unique $p_i$ and a unique $q_j$ where $\mathbf{x}^l \models p_i$ and $\mathbf{x}^l \models q_j$. Subsequently:

$$
\begin{aligned}
Pr_{n_1}(\mathbf{x}) \cdot Pr_{n_2}(\mathbf{x}) &= \Big( Pr_{p_i}(\mathbf{x}^l) \cdot Pr_{s_i}(\mathbf{x}^r) \cdot \alpha_i \Big) \cdot \Big( Pr_{q_j}(\mathbf{x}^l) \cdot Pr_{r_j}(\mathbf{x}^r) \cdot \beta_j \Big) \\
&= (Pr_{p_i}(\mathbf{x}^l) \cdot Pr_{q_j}(\mathbf{x}^l)) \cdot (Pr_{s_i}(\mathbf{x}^r) \cdot Pr_{r_j}(\mathbf{x}^r)) \cdot (\alpha_i \cdot \beta_j) \\
&= \Big( \frac{1}{\kappa_{p_i q_j}} \cdot Pr_{p_i}(\mathbf{x}^l) \cdot Pr_{q_j}(\mathbf{x}^l) \Big) \cdot \Big( \frac{1}{\kappa_{s_i r_j}} \cdot Pr_{s_i}(\mathbf{x}^r) \cdot Pr_{r_j}(\mathbf{x}^r) \Big) \cdot \Big( \kappa_{p_i q_j} \cdot \kappa_{s_i r_j} \cdot \alpha_i \cdot \beta_j \Big)
\end{aligned}
$$

where $\kappa_{p_i q_j}$ and $\kappa_{s_i r_j}$ are the normalizing constants of $p_i \cdot q_j$ and $s_i \cdot r_j$, respectively. Let $\kappa = \sum_{ij} \kappa_{p_i q_j} \cdot \kappa_{s_i r_j} \cdot \alpha_i \cdot \beta_j$ denote the normalizing constant of $Pr_{n_1} \cdot Pr_{n_2}$. The above expression corresponds to a PSDD for the desired distribution $\frac{1}{\kappa} \cdot Pr_{n_1} \cdot Pr_{n_2}$, which has elements:

$$
\Big( \frac{1}{\kappa_{p_i q_j}} \cdot p_i \cdot q_j, \ \frac{1}{\kappa_{s_i r_j}} \cdot s_i \cdot r_j, \ \frac{1}{\kappa} \cdot \kappa_{p_i q_j} \cdot \kappa_{s_i r_j} \cdot \alpha_i \cdot \beta_j \Big)
$$

Algorithm 1 recursively constructs this PSDD. The base case used in Line 3 is as follows:

| $n_1 \backslash n_2$ | $\bot$ | $X$ | $\neg X$ | $X : \beta$ |
|---|---|---|---|---|
| $\bot$ | $(\bot, 0)$ | $(\bot, 0)$ | $(\bot, 0)$ | $(\bot, 0)$ |
| $X$ | $(\bot, 0)$ | $(X, 1)$ | $(\bot, 0)$ | $(X, \beta)$ |
| $\neg X$ | $(\bot, 0)$ | $(\bot, 0)$ | $(\neg X, 1)$ | $(\neg X, 1 - \beta)$ |
| $X : \alpha$ | $(\bot, 0)$ | $(X, \alpha)$ | $(\neg X, 1 - \alpha)$ | $(X : \frac{\alpha \cdot \beta}{\kappa}, \kappa)$ |

The notation $X : \theta$ represents a simple or-gate over variable $X$ such that the literal $X$ has weight $\theta$ and the literal $\neg X$ has weight $1 - \theta$. Moreover, $\kappa = \alpha \cdot \beta + (1 - \alpha) \cdot (1 - \beta)$.

Let $i_v$ denote a node in input PSDD $n_1$ normalized for vtree node $v$ and let $j_v$ denote a node in input PSDD $n_2$ normalized for vtree $v$. Let $size(i_v)$ be the number of elements for PSDD node $i_v$ (and similarly for $j_v$). The size of input PSDD $n_1$ is then $s_1 = \sum_{v,i_v} size(i_v)$ (and similarly for $s_2$). Due to caching, we invoke the algorithm at most once for each pair of PSDD nodes $(i_v, j_v)$. Moreover, given the Cartesian product on the elements of $i_v$ and $j_v$, the overall complexity of the algorithm is

$$O(\sum_v \sum_{i_v j_v} size(i_v) \cdot size(j_v)) = O\left(\left[\sum_{v,i_v} size(i_v)\right]\left[\sum_{v,j_v} size(j_v)\right]\right) = O(s_1 s_2).$$

$\square$

**Proof of Theorem 4** We first observe that for a d-DNNF circuit $f$, checking the validity of $\exists X f$ is NP-hard [Darwiche and Marquis, 2002]. We will now reduce this test to summing out a variable from a deterministic and decomposable AC. We first convert the d-DNNF circuit $f$ into a decomposable and deterministic arithmetic circuit $\mathcal{AC}$ as given in the proof of Theorem 1. Recall that the resulting AC is such that $\mathcal{AC}(\mathbf{x}) = 0$ iff $f$ outputs 0 on input $\mathbf{x}$. Let $\mathbf{Y} = \mathbf{X} \setminus X$. Then $\exists X f$ outputs 0 on input $\mathbf{y}$ iff $(\sum_X \mathcal{AC})(\mathbf{y}) = 0$. Therefore, $\exists X f$ is valid iff $\min_{\mathbf{y}}(\sum_X \mathcal{AC})(\mathbf{y}) > 0$, which can be decided in polytime if $\sum_X \mathcal{AC}$ is decomposable and deterministic. $\square$

**Proof of Theorem 5** The proof is constructive. First, we identify a distribution that has no compact PSDD representation for any vtree. We then show that this distribution results from summing out a variable from another distribution that can be represented compactly as a PSDD.

Let PSDDs $n_1$ and $n_2$ represent two fully-factorized distributions $Pr_1$ and $Pr_2$ over variables $\mathbf{Z}$. Let PSDD $a$ represent the weighted addition $Pr_a = \theta_1 Pr_1 + \theta_2 Pr_2$ where $\theta_1$ and $\theta_2$ are positive weights that sum to one. Let $\mathbf{X}$ and $\mathbf{Y}$ be a partition of variables $\mathbf{Z}$ and consider the conditional distribution $Pr_a(\mathbf{Y} \mid \mathbf{x})$ for some instantiation $\mathbf{x}$. We now have

$$Pr_a(\mathbf{Y}, \mathbf{x}) = \theta_1 Pr_1(\mathbf{Y}, \mathbf{x}) + \theta_2 Pr_2(\mathbf{Y}, \mathbf{x}) = \theta_1 Pr_1(\mathbf{x}) Pr_1(\mathbf{Y}) + \theta_2 Pr_2(\mathbf{x}) Pr_2(\mathbf{Y})$$

$$Pr_a(\mathbf{x}) = \sum_{\mathbf{y}} Pr_a(\mathbf{xy}) = \sum_{\mathbf{y}} \left[\theta_1 Pr_1(\mathbf{xy}) + \theta_2 Pr_2(\mathbf{xy})\right] = \theta_1 Pr(\mathbf{x}) + \theta_2 Pr(\mathbf{x}).$$

Hence,

$$Pr_a(\mathbf{Y} \mid \mathbf{x}) = \frac{Pr_a(\mathbf{Y}, \mathbf{x})}{Pr_a(\mathbf{x})} = \frac{\theta_1 Pr_1(\mathbf{x}) Pr_1(\mathbf{Y}) + \theta_2 Pr_2(\mathbf{x}) Pr_2(\mathbf{Y})}{\theta_1 Pr_1(\mathbf{x}) + \theta_2 Pr_2(\mathbf{x})}$$
$$= \tau_{\mathbf{x}} Pr_1(\mathbf{Y}) + (1 - \tau_{\mathbf{x}}) Pr_2(\mathbf{Y})$$

where

$$\tau_{\mathbf{x}} = \frac{\theta_1 Pr_1(\mathbf{x})}{\theta_1 Pr_1(\mathbf{x}) + \theta_2 Pr_2(\mathbf{x})} = \frac{1}{1 + \frac{\theta_2 Pr_2(\mathbf{x})}{\theta_1 Pr_1(\mathbf{x})}}.$$

The conditional distribution $Pr_a(\mathbf{Y} \mid \mathbf{x})$ is then a weighted sum of the fully factorized distributions $Pr_1(\mathbf{Y})$ and $Pr_2(\mathbf{Y})$, where the weight $\tau_{\mathbf{x}}$ is a function of the instantiation $\mathbf{x}$. Assume that $Pr_1(\mathbf{Y})$ and $Pr_2(\mathbf{Y})$ are distinct. First, note that any distinct weight $\tau_{\mathbf{x}}$ yields a distinct conditional distribution $Pr_a(\mathbf{Y} \mid \mathbf{x})$. Second, with the appropriate parameterization of $Pr_1, Pr_2$, we can guarantee that the weight $\tau_{\mathbf{x}}$ is distinct for all distinct instantiations $\mathbf{x}$.[9] Since we have $2^{|\mathbf{X}|}$ distinct instantiations $\mathbf{x}$, we have $2^{|\mathbf{X}|}$ distinct conditional distributions $Pr_a(\mathbf{Y} \mid \mathbf{x})$.

A vtree node $v_i$ on the right most path will partition variables $\mathbf{Z}$ into $\mathbf{X}$ and $\mathbf{Y}$, where $\mathbf{Y}$ are the variables inside vtree $v_i$. Any distinct conditional distribution $Pr_a(\mathbf{Y} \mid \mathbf{x})$ must have a distinct PSDD node normalized for vtree $v_i$, leading to $2^{|\mathbf{X}|}$ such nodes in the above construction. Hence, the PSDD for $Pr_a$ is exponentially large. This is analogous to the Sieling and Wegener [1993] construction and bound for OBDDs.

Consider now the distribution $Pr_c(U, \mathbf{X}, \mathbf{Y})$ with $Pr_c(u, \mathbf{x}, \mathbf{y}) = \theta_1 Pr_1(\mathbf{X}, \mathbf{Y})$ and $Pr_c(\overline{u}, \mathbf{x}, \mathbf{y}) = \theta_2 Pr_2(\mathbf{X}, \mathbf{Y})$, where $Pr_1, Pr_2$ and $\theta_1, \theta_2$ are as given above. Distribution $Pr_c$ can be represented as a PSDD whose size is linear in $n = |\mathbf{Z}|$. Summing-out variable $U$ from $Pr_c$ results in the distribution $Pr_a(\mathbf{X}, \mathbf{Y})$, which has an exponentially large PSDD for any vtree (as shown above). $\square$

**Proof of Theorem 6** The proof (sketch) is based on constructing a particular vtree, leading to a PSDD with the mentioned size.

Assume first that we have a jointree for the given factors, whose largest cluster has size $\leq w + 1$ (such a jointree must exist by the definitions of jointree and treewidth). Designate an arbitrary cluster of the jointree as root. We will now construct a vtree recursively from this jointree/root pair.

1. The base case is for a jointree with only one cluster $\mathbf{C}$, for which we construct a right-linear vtree over the variables of $\mathbf{C}$. A right-linear vtree is a vtree in which the left child of each internal node is a leaf (e.g., the third vtree of Figure 4).

2. For the inductive case, let $\mathbf{C}$ be the jointree root.

   (a) Remove $\mathbf{C}$ from the jointree, leading to a number of disconnected trees $t_i$, while selecting the neighbor of $\mathbf{C}$ in each $t_i$ as the root for $t_i$.
   (b) Recursively construct a vtree $v_i$ from each tree $t_i$ and its root.
   (c) Connect vtree nodes $v_i$ arbitrarily into a vtree $v_{\mathbf{C}}$.
   (d) Connect vtree node $v_{\mathbf{C}}$ and the variables of cluster $\mathbf{C}$ into a right-linear vtree structure, with the variables of cluster $\mathbf{C}$ appearing first and node $v_{\mathbf{C}}$ appearing last.

This construction leads to a decision vtree as defined in [Oztok and Darwiche, 2014], where factors play the role of CNF clauses.

Suppose now that we construct a PSDD for the given factors using the above vtree and the method proposed in this paper, and let $Pr(.)$ be the distribution induced by this PSDD. Suppose that vtree node $v$ was added when processing cluster $\mathbf{C}$ (in Steps 1, 2c or 2d), where $\mathbf{X}$ are the variables of vtree $v$. One can show that we have at most $2^{|\mathbf{C}|}$ distinct distributions of the form $Pr(\mathbf{X}|\mathbf{c})$ [Kisa et al., 2014]. Hence, the PSDD will have at most $2^{|\mathbf{C}|}$ distinct nodes normalized for $v$. Moreover, one can show that each PSDD node will have two elements [Oztok and Darwiche, 2014]. Hence, the size of resulting PSDD will be $O(m \cdot \exp(w))$. $\square$

## Footnotes

[9] Note $\frac{Pr_2(\mathbf{x})}{Pr_1(\mathbf{x})} = \prod_{i \in I_{\mathbf{x}}} \frac{q_i}{p_i} \prod_{j \notin I_{\mathbf{x}}} \frac{1-q_j}{1-p_j}$ where $I_{\mathbf{x}}$ is the set of indices $i$ where $X_i$ is set to true by $\mathbf{x}$. Each $I_{\mathbf{x}}$ is unique for each distinct $\mathbf{x}$, so each $\tau_{\mathbf{x}}$ is unique if we set $\frac{1-p_i}{1-q_i} = \frac{1}{2}$ and $\frac{p_i}{q_i}$ to a unique prime for all $i$.