[Reviews · NeurIPS 2016]

Reviewer 1

Summary

The paper draws interesting and novel connections between arithmetic circuits and PSDDs and provides a groundbreaking new way to directly compile ACs for graphical models simply by multiplying PSDDs for the graphical model factors. Compilation sizes and times of this PSDD approach vs. existing compilers further indicate the benefits of this novel, much simplified method for AC construction. The paper is exceptionally clear and should be accepted in my opinion; I have only very minor comments.

Qualitative Assessment

Title: is this really the best description of the paper? It seems to me that something more like "Efficient AC Compilation of Graphical Models via PSDD Multiplication" would more clearly indicate some of the critical connections and contributions made in the paper. Question on naming: In the spirit of the relation between BDDs and ADDs, are PSDDs just Algebraic generalizations of SDDs (ASDDs) whose numerical evaluations sum to one? The later discussion of factors vs. probabilities indicates this. Then I might suggest ASDD would be a more general terminology. Theorem 1: are both decomposability and determinism required for efficient inference (presumably linear time) in the size of an AC? If so, can the authors briefly explain why inference is less tractable without these properties? My (perhaps incorrect) belief was that any AC representation of a graphical model was sufficient for linear time inference in the size of the AC. If my belief is correct, then is there a trivial PSDD multiply algorithm that does not retain these properties yet still leads to efficient inference in the result? Algorithm 1, line 19: how is this cache implemented? Like ADDs, is it based on the assumption that all subcircuits are recursively cached such that a unique ID can be maintained for all PSDDs? Whether this is the case or not, it seems worth explaining as this is critical to efficiently implementing the Multiply algorithm. Section 6: Based on the discussion of AOMDDs, its hard for me to see the difference between PSDDs and AOMDDs... can the authors elaborate more on the differences?

Confidence in this Review

3-Expert (read the paper in detail, know the area, quite certain of my opinion)


Reviewer 2

Summary

This paper presents an algorithm for compiling a Markov network into the PSDD representation of probability distributions that was introduced by Kisa et al (2014). Experiments show that this algorithm can sometimes produce smaller arithmetic circuits than existing compilers.

Qualitative Assessment

The novelty is relatively low since the compilation algorithm presented here is very similar to the compilation algorithm for AND/OR Multi-Valued Decision Diagrams (AOMDDs), which are a special case of PSDDs. Theorem 6 follows directly from the similar theorem that already holds for AOMDDs. The multiplication algorithm in section 3 is essentially the same as the one for SDDs, and it is no surprise that it operates in polytime. The main novelty and significance is in the experimental results, which suggest that PSDD compilation is more effective than AOMDD compilation. The paper would be more interesting if it gave a deeper analysis of where these advantages come from. The paper glosses over the fact that Theorem 6 does not actually apply to the compilation algorithm. Theorem 6 only says that a small PSDD exists, not that it will be found. By contrast, the corresponding theorem for AOMDDs guarantees that the compiled representation will be small. This paper will have a limited audience since it is only understandable if you have read Darwiche (2011) and Kisa et al (2014). For example, I cannot make any sense of definition 6. If each vtree node produces a single SDD ("the SDD"), why is the left child described as having multiple SDDs? It appears to be an adaptation of definition 5 from Darwiche (2011), but that definition is clear while this definition is not. I suggest having another go at writing this definition, or simply copy definition 5 from Darwiche. When discussing the relationship between SDD and PSDD, it is important to point out (as Kisa did) that the SDD underlying a PSDD is not compressed. Line 111 of the paper seems to suggest the opposite. There should be a more extensive discussion of the relationship between PSDDs and AOMDDs, especially the fact that AOMDDs are a special case. This implies that PSDDs are more succinct but potentially less tractable than AOMDDs, in the terminology of Darwiche (2011). This seems to be illustrated by the fact that the compilation algorithm for AOMDDs is provably upper bounded by treewidth while the same algorithm for PSDDs is not.

Confidence in this Review

2-Confident (read it all; understood it all reasonably well)


Reviewer 3

Summary

This work proposes to use PSDDs instead of factor and model counting based representations. It is shown that multiplication can be performed efficiently which leads to an efficient algorithm to represent the AC that represents the distribution of a PGM, which is a product of factors.

Qualitative Assessment

This work offers an interesting and useful new approach to inference in probabilistic models. The work is explained clearly and in sufficient in detail. Although the experiments show progress I am not entirely sure how big of an impact this method would have (in contrast with being another heuristic/approach to represent compact ACs that often show large variance of performance on different problems). But looking at other work that reduced auxiliary variables when performing knowledge compilation for PGMs also has shown the benefits of having less variables, I am inclined to believe this can have a large impact. Detailed comments: - Alg 1, line 12+13: multiply returns a tuple, should "P,c1" and "S,c2" and the cache calls in the line 15 can be replaced with c1 and c2. - Sec 4, theorem 4: I miss the intuition (or ref) to really understand this theorem. - Sec 4, line 212+213: "the product AC". Should this be 'resulting' AC? Minor: - Sec 3, line 156: Fix citation format.

Confidence in this Review

3-Expert (read the paper in detail, know the area, quite certain of my opinion)


Reviewer 4

Summary

The authors describe algorithms for operations on arithmetic circuits suitable for representing probability distributions. Such circuits can be used as a sparse and efficient representation for probability distributions, therefore the manuscript has high potential impact for practical applications in probabilistic reasoning and fits NIPS. Concretely the authors work on PSDDs (probabilistic sentential decision diagrams) and describe theoretical results on multiplying such representations, compiling PSDDs into arithmetic circuits, multiplying two PSDDs, summing out probabilities, several representation results and an empirical comparison with other approaches.

Qualitative Assessment

The manuscript advances the theoretical framework of PSDDs and contains contributions that can make applications on probabilistic representations more efficient in the future. The only drawback of the manuscript is, that it is not self- contained and requires background reading to intuitively understand, because the example (figures) are not explained in detailed and because some preliminaries are not given. Nevertheless I recommend acceptance for publication. Detailed drawbacks of examples: * Figure 1 shows a circuit for a network A -> B and it contains numbers attached to nodes, however it is not clear what these numbers are, and how they are related to A -> B. Also the text does not explain this. From my interpretation, this is just an arbitrary example situation, where \theta_a = 0.3, ... Curiously, \lambda_b = 1 and \lambda_\not{b} = 1 although these should be inverse if they are indicators. It does not help that Figure 1 and Figure 2 seem to describe two complete distinct examples (Fig 2 has 3 variables, Fig 1 has only two). * Figure 2 shows several numbers (1, 1, 3, 4) that are not intuitive. In (b) and (c) they are attached to the topologically same or-gates, however in (d) in addition there is "0" and "2", and only one "1". I recommend to revise the presentation of these examples, putting the figures later (to the text that describes it) and perhaps putting the figures on separate pages because they seem to be unrelated. (I wonder if the and/or gate way of drawing PSDDs is a good choice, as drawing them just as graphs with edge labels would be more close to the original ACs, however this seems to be a choice of the authors of previous papers.) Minor comments: * 18: compilating -> compiling * Figure 4: top-aligning the three trees could be nicer * Table 1: winning (shortest) times could be marked in bold * Table 1: what is the unit of size? not MB as it seems. nodes?

Confidence in this Review

2-Confident (read it all; understood it all reasonably well)


Reviewer 5

Summary

There is a collection of small results in the paper including intractability of multiplication and summing-out (i.e. marginalization) for arithmetic circuits (ACs) with certain properties. A recently defined subclass of ACs, called PSDDs has been defined in the paper. The main results in the paper concern PSDDs. The two main contributions include providing an algorithm for multiplication of PSDDs and showing that it can be done in polynomial time (which is a limitation of general tractable circuits); and providing a corresponding arithmetic circuit that is both decomposable and deterministic. These lead to new method for compiling probabilistic graphical models, which the authors compare with the existing ones in this paper.

Qualitative Assessment

The paper provides simple results and algorithms that, to the best of my knowledge, are quite clever and novel. However, the resulting method seems to marginally contribute to the literature. One of my concerns is that the algorithm for multiplication of PSDDs is not very well-described, and in particular there are lines in the middle of the algorithm that are unclear. I understand the space limitation, but at least expected to see some mention of this in the supplementary material. A similar concern is for the comparison with other models in the literature. It seems as though there is a significant difference on how well the newly defined method works based on the choice of the network. There should be at least a discussion on for what types of networks this method provides a significant improvement. There is also virtually no mention of the compilation time, which does not seem to have improved at all with this new method (which was rather surprising to me). The quality of writing of the paper is also not very high. There are repeated mistakes here and there. Two of the major ones: missing or existing unnecessary "s" at the end of verbs (e.g. p3 l105 (admits) or p7 l215 (exist) etc.). There is also a confusion with references in the text; e.g. p5 l155, it should be "[Darwiche, 2009]", but on l161 it should be "Larkin and Dechter [2003]". A question: do you not think the projection discussed graphically (e.g. Figure 4) works in the same way as marginalization over the remaining nodes and their parents in DAGs?

Confidence in this Review

2-Confident (read it all; understood it all reasonably well)